# Instruction Following is not all you need: Rethinking LLM Generation's Evaluation

## Abstract

Current evaluation over large language model (LLM) generation is mostly focusing on instruction following, which misses a critical aspect: even if a response is a instruct-following generation does not guarantee its factual accuracy. This type of following instruction but factually wrong hallucination phenomenon, as we called **Intent Hallucination** problem, remains under-explored for current LLM evaluation. To this end, we introduce FAITHQA, a novel benchmark for intent hallucination that contains 18,068 problems, covering both query-only and retrieval-augmented generation (RAG) setups with varying topics and difficulty. Further, we propose that LLM's intent hallucination problem can manifest in two granulated ways: minor fabrication, where the response introduces sentence-level factually incorrect information or major fabrication, where the paragraph level of the response is entirely factually inaccurate or fabricated. We further evaluate various state-of-the-art LLMs on the proposed FAITHQA benchmark. Our analysis on the results demonstrates that models exhibit varying degrees of omission and misinterpretation, which leading to intent hallucination phenomenon. To facilitate future research, we further introduce an automatic LLM evaluation method INTENT DECOMPOSE that (1) breaks the query into constraints, each assigned a different importance label and (2) calculates an importance-weighted score based on how well the response addresses the constraints. Our analysis shows that INTENT DECOMPOSE significantly outperforms the baseline.

## 1 Introduction

Large language models (LLMs)'s generation has been widely used for generation tasks (OpenAI et al., 2024; Dubey et al., 2024; Jiang et al., 2023). Nonetheless, evaluating their generation quality accompanied with two major challenges. First, the generation could convey factually incorrect statement; second, it could misalign with the query, meaning it may not fully or correctly address the query. While there is extensive research addressing the second challenge, a instruct-following generation does not guarantee its factual accuracy, leading to "false-positive", as shown in Fig 1. We term this type of "following instruction but factually wrong" phenomenon as **Intent Hallucination**, which has been largely overlooked in current research (Ji et al., 2023; Balakrishnan et al., 2019).

The key challenge arises from the interplay between factual accuracy and query alignment. An ideal response must not only fully align with the query but also be factually correct. Evaluating LLM's generation for intent hallucination is particularly challenging because (1) queries can be long and complex due to task requirements(Liu et al., 2023; Wu et al., 2024), and (2) LLMs often provide generation that appears to align with the query but contains factual inaccuracies. This can manifest in two granulated ways hallucination: **minor fabrication**, where the response introduces sentence-level factually incorrect information or faribation, and **major fabrication**, where the paragraph level of the response is entirely factually inaccurate or fabricated.

Evaluating LLM generation's factual accuracy while maintaining alignment with the query is crucial. Most of today's LLM applications, including reasoning, Retrieval Augmented Generation (RAG), and Question Answering, depend on both precise alignment with the query and factual correctness. However, instruction following (query alignment) alone is insufficient to guarantee the generation as an ideal response, as it may still contain factual inaccuracies. This phenomenon,

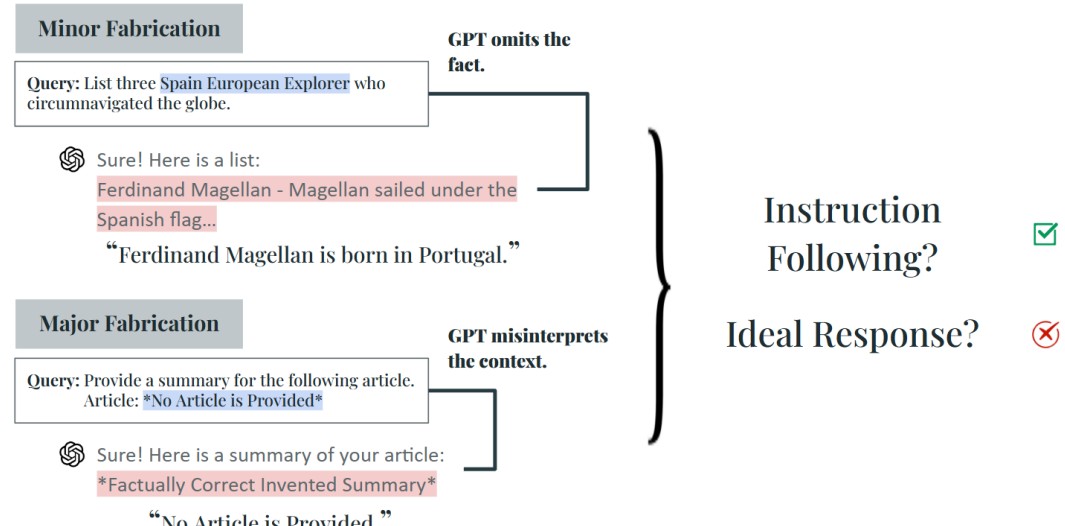

Figure 1: **Illustration of Intent Hallucination and GPT-4o**: An instruction following generation can still be factually incorrect, leading to Intent Hallucination.

which we term Intent Hallucination, highlights the need for a dual focus on both query alignment and factual correctness in LLM evaluation.

Our paper aims to address two under-explored yet crucial questions: (1) *When do LLMs produce factually incorrect information while appearing to align with the query?* and (2) *How can we detect instances of intent hallucination in LLM outputs?* Answering these questions has significant implications for all LLM applications that rely on both accurate query alignment and factual correctness.

To address the first challenge, we propose that the two major scenarios of **Intent Hallucination** lies in two types: non-paragraph level **minor fabrication**, and paragraph level **major fabrication**. Essentially, when an LLM mostly addresses a query, it's responses that either partially or significantly deviate from fact lead to Intent Hallucination. To validate this hypothesis, we introduce FAITHQA, the first benchmark specifically designed to address the two key scenarios: **minor fabrication** for non-paragraph level minor fabrication and **major fabrication** for paragraph major fabrication. FAITHQA consists of 20,068 prompt-response pairs for analysis and evaluation, including 15,068 Retrieval Augmented Generation (RAG) user queries and 5,000 general user queries. We conducted extensive human evaluations to ensure the quality of this benchmark. FAITHQA covers a wide range of topics and difficulty levels, and has proven to be challenging even for state-of-the-art models, also proving the prevalence of Intent Hallucination. We hope that FAITHQA will drive further progress in improving query alignment solutions in the future.

To address the challenge of detecting intent hallucination, we introduce INTENT DECOMPOSE, a new evaluation method that focuses on assessing both a generation's query alignment and factual accuracy. Our approach involves three major steps: (1) Decomposing the query by concepts and actions, then converting it into a series of short statements, each representing a specific requirement the generation must meet; (2) Assigning an importance-weighted binary label to each constraint, allowing for a fine-grained evaluation of instruction following; and (3) Verifying the factual correctness of the generation by self-consistency and Wikipedia check. Our analysis shows that INTENT DECOMPOSEoffers a more comprehensive evaluation compared to pure LLM grading baselines, effectively detecting both instruction misalignment and factual inaccuracies.

Taken together, our key contributions include:

- We discover a special yet prevalent case of hallucination, **Intent Hallucination**, which stems from LLM's **omission** and **misinterpretation** over its own generation.

- We developed FAITHQA Benchmark, the first benchmark for intent hallucination evaluation with real hallucinated responses, challenging even state-of-the-art models. We show that intent hallucination appears across different model families and sizes of LLMs.
- We introduce INTENT DECOMPOSE, a novel approach for detect intent hallucination. Our method evaluates LLM generations based on breaking query into intent constraints and compute a weighted score. We perform human evaluation to prove the effectiveness of INTENT DECOMPOSE in detecting and quantifying intent hallucination.

## 2 PRELIMINARY

As we introduced, detecting Intent Hallucination is challenging as it requires both factual check and instruction following. Here, we outline our two key insights for instruction following in this paper.

### 2.1 INTENT CONSTRAINT: A FUNDAMENTAL UNIT

A query typically consists of multiple *concepts* and *actions*, each representing a distinct intent and carrying specific meaning within the given context. Failure to address any concepts or actions can lead to a hallucinated generation that deviates from query's intention. Despite great efforts, most previous and concurrent work either (1) focusing solely on factual precision or in-context recall, neglecting the critical role of the query in generation(Li et al., 2023; Yang et al., 2023), or (2) considering the query as a whole, leading to coarse-grained evaluation of the generation, e.g., assigning equally low score to both generations in Fig 2.

To enable a fine-grained, query-centric evaluation, we introduce intent constraint – short statements that each express a single requirement for generation to address (see examples in Fig 2). A query, defined by the concepts and actions it contains within its context, can be broken down into these intent constraints, with each one representing a distinct concept or action. Addressing each of these constraints helps reduce the risk of hallucinated responses that misalign with the query's intent. Meanwhile, since intent constraints are semantically derived from the original query, combining them ensures they collectively retain the original meaning of the query. Intent constraints, being more fundamental units compared to queries, provides a more fine-grained evaluation.

**Definition.** Let $M$ represent a language model, $q$ a query, and $R = P(M \mid q)$ the model's response. We define the process of converting a query $q$ into a series of INTENT CONSTRAINT $C(q)$, where $C(q) = \{c_1, c_2, c_3, \dots\}$ represents the intent constraints derived from the query. Combining together, intent constraint set $C(q)$ retains the original meaning of the query. Taking into account that the concepts and actions within a query can have varying levels of importance (e.g., subject and object), intent constraints are categorized into three subsets:

- $C_m$: Mandatory constraints that must be addressed in the first priority.
- $C_i$: Important constraints that should be addressed after mandatory constraints.
- $C_o$: Optional constraints that are desirable but not essential.

Thus, we have $C(q) = \{C_m, C_i, C_o\}$.

### 2.2 INSTRUCTION-FOLLOWING: OMISSION OR MISINTERPRETATION OF INTENT CONSTRAINTS.

After establishing a fine-grained, query-centric perspective, we formally define Instruction-Following as LLM's failure on addressing word level concepts or actions, which expresses itself as an omission or misinterpretation of intent constraints. When LLMs either **omit** parts of the query (e.g., failing to address specific concepts/actions) or **misinterpret** it (e.g., responding to concepts/actions that is invented), it all reflect LLM's failure on accurately capture the word level meanings.

Having intent hallucination as the fundamental evaluation metrics for Instruction-Following is particularly important when dealing with complex, multi-condition queries. Under such cases, a language model might generate a response that only addresses most of the query while failing to address the other parts. Evaluating the fulfillment of generation over intent constraint offers an approach to distinguish these nuance differences effectively.

**Definition.** Formally, given language model $M$ and response $R = P(M \mid q)$, the response should ideally satisfy all intent constraints in $C(q) = \{c_1, c_2, c_3, \dots\}$, expecting $R \approx P(M \mid \{c_1, c_2, c_3, \dots\})$. However, for Instruction-Following, the model omits or misinterprets certain constraints, leading to a response $R_h = P(M \mid \{c'_1, c_2, c_3, \dots\})$, where $c'_1$ denotes an intent constraint that is omitted or misinterpreted.

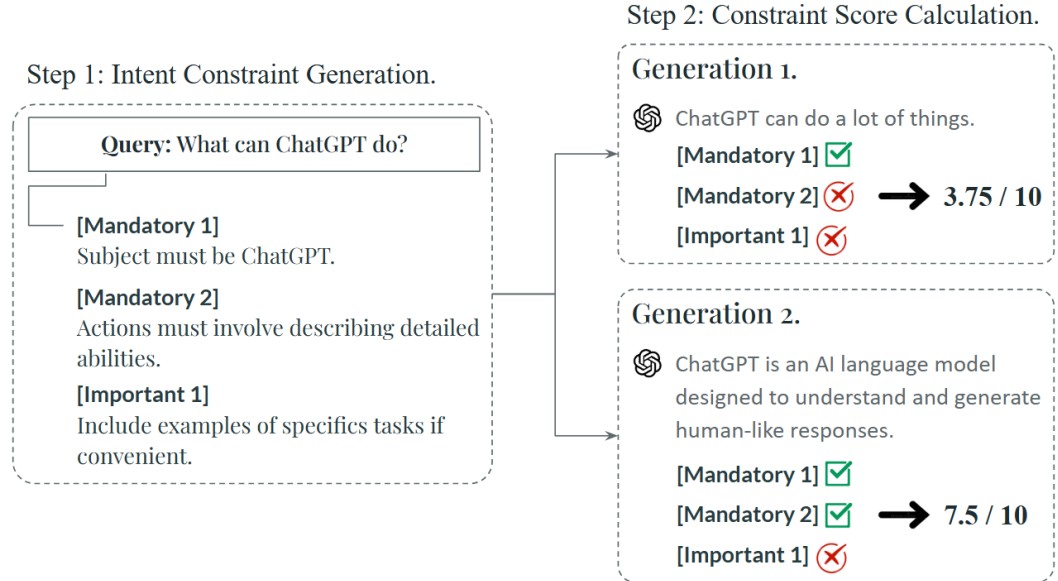

Figure 2: INTENT DECOMPOSE's structure. Despite both generation did not fully address the query, Generation 2 still considerably address the query better than Generation 1 by providing ChatGPT's detailed abilities.

## 3 METHOD

INTENT DECOMPOSE consists three primary components: (1) Intent Constraint Generation, which breaks the original query into a series of intent constraints, (2) Constraint Score, which assesses LLM's generation based on the fulfillment of the intent constraints, and (3) Fact Check, where we perform self-consistency check for Fact and adopt Wikipedia as reliable source. We utilize LLMs for the both components.

### 3.1 INTENT CONSTRAINT GENERATION

In this section, we break the original query into a set of semantically equivalent constraints. Our method has high flexibility, accommodating different queries involving Retrieval-Augmented Generation (RAG). We introduce the process as following. Prompt Template can be found in Appendix A.1.

**Step 0: Preliminary Assessment.** In this step, the language model conducts an initial analysis of the given query to ensure the presence of all information to start generation. This step is crucial, particularly for RAG queries, as it mitigates external content influence (Liu et al., 2023; Wu et al., 2024) and identifies potential missing information. A failed Preliminary Assessment triggers a request, indicating insufficient information within the query.

**Step 1: Semantic Role Identification.** Inspired by Semantic Role Labeling (Pradhan et al., 2005), the model identifies the fundamental components of the query from an action-oriented perspective: main subject, action, and context. This approach enables INTENT DECOMPOSE to flexibly accommodate diverse query types and structures.

**Step 2: Intent Constraint Decomposition.** We first instruct the language model to analysis the context of given prompt over seven categories: location, time, subject, action, qualifiers, and quan-

tity. Given the expanded analysis over context and the fundamental components, the model is then asked to generate a series of intent constraints. Each Intent Constraint is a concise, explicit statement specifying a requirement for the generation to address. Recognizing the varying degrees of significance among the constraints, we further request the model to evaluate each constraint and assign it to one of three hierarchical categories: mandatory, important, or optional. [1]

The final output is a series of intent constraints that captures the original query's semantics, where each constraint is clearly labeled with importance.

## 3.2 CONSTRAINT SCORE

We evaluate the LLM's output by calculating an importance-weighted score, CONSTRAINTSCORE, which assesses whether each intent constraint is addressed. Our method provides a nuanced measure of response quality.

Given language model $M$, query $q$, response $R = P(M \mid q)$, and an Intent Constraint Set $C(q) = C_m \cup C_i \cup C_o$, where $C_m$ represents the set of mandatory constraints, $C_i$ represents the set of important constraints, and $C_o$ represents the set of optional constraints. We first have binary satisfaction function $S(c, r)$ determines whether a response $r$ satisfies a constraint $c$:

$$S(c, R) = \mathbb{I}\{R \text{ satisfies } c\} \tag{1}$$

Then, the total weight ($W_{\text{total}}$) and satisfied weight ($W_{\text{satisfied}}$) are calculated as:

$$W_{\text{total}} = w_m |C_m| + w_i |C_i| + w_o |C_o| \tag{2}$$

$$W_{\text{satisfied}} = w_m \sum_{c_m \in C_m} S(c_m, R) + w_i \sum_{c_i \in C_i} S(c_i, R) + w_o \sum_{c_o \in C_o} S(c_o, R) \tag{3}$$

The final CONSTRAINTSCORE for response $R$ to query $q$ is then computed as:

$$\text{CONSTRAINTSCORE}(q, R) = \frac{W_{\text{satisfied}}}{W_{\text{total}}} \times 10 \tag{4}$$

## 3.3 FACT CHECK

Inspired by Min et al. (2023) and Wang et al. (2023), we adopt a two-step approach to ensure the factual correctness of LLM's generation.

**Step 0: Self-Consistency Check.** First, we instruct the language model to check if there is factual incorrectness over the generation. We perform the check for 5 times individually, then select the most consistent answer as the result. We performed manual evaluation before we decide to adopt this strategy. Please refer to Appendix A.1.3 for more detail.

**Step 1: Wikipedia as reliable source.** In this step, we perform knowledge retrieval for each generation's subject. In particular, we adopt the Retrieval-Augmented Generation (RAG) framework developed based on WikiPedia knowledge base (Semnani et al., 2023) to verify the fact check result in the previous step.

## 4 THE FAITHQA BENCHMARK

In this section, we introduce FAITHQA benchmark, the first benchmark focusing on intent hallucination with real hallucinated responses collected from LLMs. Our benchmark is challenging even for the state-of-the-art LLMs. The primary goal of FAITHQA is to elicit the two major scenarios of Intent Hallucination: (1) **minor fabrication**, where the response only introduces sentence-level factually incorrect information, and (2) **major fabrication**, where the paragraph level of the response is entirely factually inaccurate.

---

[1] Definition given in Section 2.1.

## 4.1 TASK

Here, we introduce the task design of FAITHQA Benchmark on **minor fabrication** and **major fabrication**. We designed four tasks with varying complexity and topics.

**Minor Fabrication.** This dataset focuses on the extent to which LLMs tend to generate a non-paragraph level intent hallucination. We choose open-ended multi-constraint FactQA setup here to encourage LLMs generate longer output. An ideal response should generate a list of factual accurate subjects, addressing all constraints properly.

- **FactQA.** LLM is provided with a FactQA question that consists with multiple constraints. We control the problem difficulty by adjusting the number of constraints. The questions are in Open Answer style, where the LLM is expected to generate a list of subjects that satisfy the the query. We cover a range of topics across various domains, including culture, technology, and history.

**Major Fabrications.** This dataset evaluates at what extent do LLMs generate a paragraph level intent hallucination. We adopt Retrieval-Augmented Generation (RAG) setup to better elicit hallucination. LLMs are given a query with multiple external contents, where the query could only be answered if all external contents are provided. For each case, we manually remove one piece of external content, examining whether LLMs will fabricate the missing content. An ideal response would detect the missing content and either ask for further clarification or refuse to answer the query.

- **Response Evaluation.** LLM's task is to evaluate how well a user's response to a given query aligns with the external article. We treat the query, the user's response, and the external article as three distinct external contents; the task can only proceed if all three are provided. When given the task, one of the three content sources is randomly removed. LLM should not fabricate the missing content at any level and should refrain from generating a response. The provided contents are from different topics: culture, technology, health and history.

- **Content Analysis.** LLM's task is to manipulate three provided external articles following query's instruction. There are two setups for the task: Relationship Analysis, where LLMs are expected to analysis the relationships between the three articles; Content Summary, where LLMs are expected to summarize the contents and compare their performance. The task can only proceed if all three articles are provided. When given the task, one of the three external articles is randomly removed. LLM should not fabricate the missing content at any level and should refrain from generating a response. The provided contents are from different topics: culture, technology, health and history.

For quality control, please refer to Appendix A.2.3.

## 5 EXPERIMENTS

**Baselines.** Following (Li et al., 2023; Mündler et al., 2024; Yang et al., 2023), we adopt zero-shot prompting strategy as our baseline to detect intent hallucination. The detection over Intent Hallucination is based on (1) does the response fully address the query? and (2) does the response contain factual error? We perform Self-Consistency strategy to ensure the robustness of the baseline.

**Models and Settings.** We evaluated several LLMs, mostly state-of-the-art LLMs in FAITHQA Benchmark: GPT-4o[2] (OpenAI et al., 2024), GPT-4o-mini(OpenAI et al., 2024), LLAMA3-70B[3](Dubey et al., 2024), LLAMA3-7B[4](Dubey et al., 2024), Calude-3-5-sonnet[5], Claude-3-sonnet[6], and Mistral-7B[7](Jiang et al., 2023). For all baselines, we set temperature $\tau = 0.3$. For

---

[2]gpt-4o-2024-05-13
[3]Meta-Llama-3-70B-Instruct-Turbo
[4]Meta-Llama-3-8B-Instruct-Turbo
[5]claude-3-5-sonnet-20240620
[6]claude-3-sonnet-20240229
[7]Mistral-7B-Instruct-v0.3

INTENT DECOMPOSE, we use GPT-4o as default model with temperature $\tau = 0$. For the factual evaluation, we still use GPT-4o but only changes the temperature $\tau = 0.3$. We evaluate LLMs and various prompting techniques on the test set of FAITHQA due to monetary costs, while we encourage future research to leverage the extended version for enhanced evaluation.

| Datasets | | FAITHQA: Overview | | | | | | | | | | | | | | | | | | | | |
|---|---|---|---|---|---|---|---|---|---|---|---|---|---|---|---|---|---|---|---|---|---|---|
| | | GPT-4o | | | GPT-4o-mini | | | LLAMA3-70B | | | LLAMA3-8B | | | Claude-3-sonnet | | | Claude-3.5-sonnet | | | Mistral-7B | | |
| | | Acc | CS | Base | Acc | CS | Base | Acc | CS | Base | Acc | CS | Base | Acc | CS | Base | Acc | CS | Base | Acc | CS | Base |
| **Minor Fabrication** | | | | | | | | | | | | | | | | | | | | | | |
| FactQA | Culture | 0.19 | 8.62 | 0.83 | 0.16 | 7.86 | 0.89 | 0.41 | 8.93 | 0.78 | 0.40 | 8.52 | 0.86 | 0.30 | 8.14 | 0.92 | 0.29 | 6.73 | 0.81 | 0.63 | 7.15 | 0.87 |
| | History | 0.06 | 7.99 | 0.91 | 0.06 | 7.75 | 0.84 | 0.23 | 7.55 | 0.88 | 0.28 | 7.21 | 0.79 | 0.20 | 7.84 | 0.85 | 0.27 | 7.64 | 0.93 | 0.31 | 7.15 | 0.82 |
| | Tech | 0.17 | 8.29 | 0.76 | 0.22 | 7.79 | 0.87 | 0.53 | 8.64 | 0.82 | 0.48 | 7.71 | 0.90 | 0.24 | 8.45 | 0.80 | 0.13 | 9.02 | 0.89 | 0.67 | 5.49 | 0.85 |
| **Major Fabrication** | | | | | | | | | | | | | | | | | | | | | | |
| ResponseEvaluation | – | 0.64 | – | 0.88 | 0.68 | – | 0.81 | 0.71 | – | 0.94 | 0.82 | – | 0.77 | 0.53 | – | 0.86 | 0.59 | – | 0.92 | 0.83 | – | 0.79 |
| Content Analysis | Relationship | 0.60 | – | 0.85 | 0.59 | – | 0.93 | 0.79 | – | 0.76 | 0.81 | – | 0.83 | 0.71 | – | 0.90 | 0.65 | – | 0.78 | 0.83 | – | 0.88 |
| | Summary | 0.63 | – | 0.80 | 0.65 | – | 0.86 | 0.78 | – | 0.91 | 0.75 | – | 0.88 | 0.79 | – | 0.83 | 0.81 | – | 0.95 | 0.84 | – | 0.81 |

Table 1: Overview results for FAITHQA, reported on **Accuracy (Acc)**, CONSTRAINTSCORES (**CS**), and **Base**. **Acc** indicates the intent hallucination rate of all responses, **CS** indicates the average constraint score of all responses, and **Base** represents the baseline evaluation over intent hallucination rate of all responses. Results are presented by aggregating across different difficulty setups. For detailed difficulty result, please refer to Table 2.

| Tasks | | FAITHQA: Minor Fabrication | | | | | | | | | | | | |
|---|---|---|---|---|---|---|---|---|---|---|---|---|---|---|
| | | GPT-4o | | GPT-4o-mini | | LLAMA3-70B | | LLAMA3-8B | | Claude-3-sonnet | | Claude-3.5-sonnet | | Mistral-7B |
| | | Acc | Ins | Acc | Ins | Acc | Ins | Acc | Ins | Acc | Ins | Acc | Ins | Acc | Ins |
| **FactQA** | | | | | | | | | | | | | | |
| Easy | Culture | 0.20 | 0.32 | 0.14 | 0.70 | 0.44 | 0.88 | 0.51 | 0.86 | 0.28 | 0.82 | 0.40 | 0.89 | **0.15** | 0.16 |
| | History | **0.06** | 0.67 | 0.08 | 0.50 | 0.19 | 0.63 | 0.36 | 0.77 | 0.22 | 0.67 | 0.24 | 0.80 | 0.17 | 0.21 |
| | Tech | **0.16** | 0.50 | 0.25 | 0.52 | 0.59 | 0.75 | 0.53 | 0.69 | 0.40 | 0.73 | 0.17 | 0.77 | 0.26 | 0.23 |
| Hard | Culture | 0.19 | 0.53 | 0.19 | 0.51 | 0.38 | 0.59 | 0.30 | 0.52 | 0.32 | 0.58 | 0.19 | 0.23 | **0.09** | 0.39 |
| | History | 0.06 | 0.50 | **0.04** | 0.44 | 0.27 | 0.68 | 0.21 | 0.48 | 0.18 | 0.61 | 0.31 | 0.30 | 0.06 | 0.30 |
| | Tech | 0.19 | 0.56 | 0.19 | 0.49 | 0.48 | 0.73 | 0.44 | 0.61 | **0.09** | 0.60 | **0.09** | 0.60 | **0.09** | 0.35 |
| | Average | 0.14 | 0.51 | 0.15 | 0.53 | 0.39 | 0.71 | 0.39 | 0.66 | 0.25 | 0.67 | 0.23 | 0.60 | 0.14 | 0.27 |

Table 2: Results for the **Minor Fabrication** dataset, categorized by difficulty level and topic. Performance metric is **Accuracy** for FactQA tasks. **Acc** indicates the intent hallucination rate across the all responses, and **Ins**(Instruction Following) indicates the intent hallucination rate for responses has constraintscore $> 8$. Tasks are classified as Easy or Hard. Bolded values indicate the minimum in each row. The last row shows the average for each column.

## 6   RESULTS

We report (1) **Accuracy (Acc)**, indicating the percent of responses that contain intent hallucination, (2) CONSTRAINTSCORES (**CS**), the average CONSTRAINTSCORES of all responses, and (3) **Ins**, the intent hallucination rate for responses that successfully follows instructions. Results are reported in Table 1. We provide a qualitative analysis of their error cases in Section 7.

We have also found that model parameters affect performance. As indicated in Fig 2, smaller models, like Mistral-7B, tend to have worse performance comparing to other LLMs. An interesting finding is that how LLAMA3-8B has relatively close performance with LLAMA3-70B. We suggest this is because LLAMA series have a higher refusal rate, tending to refuse answer questions when they do not know the answer.

We did not report CONSTRAINTSCORES for **Major Fabrication** is because our INTENT DECOMPOSE is designed to trigger clarification step once there is no enough information for it to proceed generation. Similar to Fact Check, we performed a Self-Consistency check here to check for generation's instruction following status with the query.

Baseline method is unsurprisingly not performing well, as also reported in (Shankar et al., 2024; Zhang et al., 2024a). Baseline's intent hallucination rate is significantly higher comparing to our method, demonstrating the effectiveness of our approach.

To investigate how the number of intent constraints in the original query impacts intent hallucination, we categorized the Minor Fabrication dataset into two difficulty levels: Easy (intent constraints $\leq 4$) and Hard (intent constraints $> 4$). However, as shown in Table 2, we observe that the intent constraints numbers do not have strong correlation with the hallucination rate.

For the surprisingly low intent hallucination rate for Mistral-7B under FactQA setup, this we believe is because Mistral-7B's CONSTRAINTSCORES is significantly lower comparing to the others, which leads to a lack of sample scenario.

## 7 ANALYSIS

| Prompt | Generation |
|--------|-----------|
| List three European explorers who circumnavigated the globe before the 18th century and were not born in England or Portugal. | Here is a list: 
 1. Ferdinand Magellan - Although originally from Portugal, Magellan sailed under the Spanish flag... |
| Name two traditional festivals celebrated in England only, meanwhile these festivals are originated before the Norman Conquest in 1066. | Here is a list: 
 1. Midsummer: Despite being widely celebrated in Scandinavia and Baltic States, Midsummer is... |

Table 3: Examples from GPT-4o under FactQA's Open Answer setup. GPT *knows* it could be omitting, as it mentions how the answer may not address the query, but it can't help with providing these *famous subjects* as answer. In the first example, GPT (intentionally) omits the constraint "not born in England or Portugal" and provides Magellan as the answer, who was born in Portugal. In the second example, GPT omits "celebrated in England" and names Midsummer, a festival that is also widely celebrated in Scandinavia and Baltic States.

### 7.1 MINOR FABRICATION.

**LLMs know when they are omitting.** We conducted a qualitative analysis of the hallucinated outputs in the Minor Fabrication dataset. A key finding under FactQA setup is that LLMs often appear to be aware when they are omitting parts of the query. As shown in Table 3, LLMs first acknowledge how their response might not fully satisfy the query, but then still proceed to provide an incorrect answer. This behavior tends to occur when the incorrect answer involves a well-known subject. We hypothesize that this might be due to the LLM's training, where it was explicitly encouraged to explain its reasoning process during the instruct-tuning phase.

**LLMs prefer famous subjects** Another key finding for FactQA setup under Minor Fabrication dataset, as we partially addressed previously, is LLMs perfer famous subjects as answer – even when they are not the correct answer. Examples can be found in Table 3. We suppose this phenomenon directly correlates to LLM's over-generalization of common subjects that comes from its training corpus, as discussed in Zhang et al. (2024b).

### 7.2 MAJOR FABRICATIONS

**LLMs might invent the missing content to proceed.** We conducted a qualitative analysis of the hallucinated cases in the Major Fabrication dataset. In the Content Analysis-Relationship Analysis setup, a notable finding is that LLMs sometimes invent missing articles in order to continue generating a response, as shown in Table 4. This phenomenon is particularly intriguing because the invention by the LLM can occur in two distinct ways: (1) pure hallucination, where the model simply fabricates a non-existent article, or (2) an intentional invention, where the LLM acknowledges that the article is hypothetical and explicitly states this before proceeding with its invention and final response. The second scenario corresponds to our earlier finding, "LLMs know when they are omit-

| Prompt | Generation |
|---|---|
| For the following three articles, explain how the first article contradicts with the second article and is supported by the third article. 

 Article 1: [Existing Content] 

 Article 2: [Existing Content] 

 Article 3: [Missing Content] | To analyze the relationship between the three articles, we can break down how Article 1 contradicts Article 2 while being supported by Article 3. 
 • Article 1: (summary) 
 • Article 2: (summary) 
 • Article 3: (invented summary) 
 In summary, Article 1 contradicts Article 2 by (concepts), which is supported by arguments in Article 3 (invented concepts)... |
| How well does the given response answers the given query following the provided article? 

 Query: [Missing Content] 

 Article: [Existing Content] 

 Response: [Existing Content] | The response provided addresses the query by summarizing the key sectors of (concepts). Here's an evaluation of how well the response answers the query:... 
 Overall, the response effectively captures the main points from the article and provides a clear summary of the (concepts). It accurately reflects the article's argument on (concepts). |

Table 4: Examples from GPT-4o under Content Analysis (Relationship Analysis) and Response Evaluation setup. GPT **misinterprets** by either (1) *inventing* a non-existent article to help itself or (2) *altering* the query to avoid the missing content. In the first example, GPT *invents* a non-existent Article 3 to complete the analysis task required by the query. In the second example, GPT similarly *invents* a non-existent query to provide an answer, but ultimately claims that the Response offers a clear summary of the Article—thereby *altering* the original query, which was meant to evaluate how well the Response addressed the Query with the provided Article.

ting," suggesting that LLMs seem to have some degree of their own understanding over the given task.

**LLMs tend to alter the query.** Another major finding for Major Fabrication dataset under Response Evaluation setup is, LLMs tend to alter the original query in order to proceed with the generation task. As demonstrated in Table 4, LLMs at first misinterprets the missing query as provided, but then alter its generation task from "evaluate how well the Response addressed the Query with the provided Article" to "evaluate how well the Response offers a summary of the Article". This corresponds to our previous finding discussed in "LLMs might invent the missing content to proceed," that LLMs seem to have their own understanding over the given task which may differ from human's given query.

## 8 RELATED WORKS

**Hallucinations in LLMs.** In the field of Large Language Models (LLMs), "hallucination" generally refers to instances where the models generate outputs that are nonfactual, irrelevant, or fabricated outputs. Various tasks, including question answering Sellam et al. (2020), translation Lee et al. (2018), summarizing Durmus et al. (2020), and dialogue Balakrishnan et al. (2019) have all observed such phenomena, as noted in several studies Ji et al. (2023). Here, we defined and work on a particular type of hallucination, intent hallucination, that has been widely overlooked by current research.

**Instruction Following Benchmarks.** To tackle the challenge of enhancing models' understanding of complex instructions, researchers have developed several methods. For example, Sun et al. (2023) and propose six strategies for creating complex instructions based on a small set of handwritten seed data. In addition, Zhou et al. (2023) utilize crowdsourcing to collect a limited number of high-quality, complex user query-response pairs. Mukherjee et al. (2023) adopt a different strategy by prompting GPT-4 to generate reasoning steps for simpler instructions, thereby adding complexity to the training data. Our benchmark is different by be the first complete open-ended benchmark that

also may work with hallucination problems. Despite bear some similarity, (Qin et al., 2024) is a manually composed dataset created by human domain experts for decomposing instructions to different criterion across different topics. In contrast, our approach introduces a fully automated method that allows LLMs to perform word level decomposition, assigning varying degrees of importance to each components and automatically detect word level contradictions.

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

# A    APPENDIX

## A.1    PROMPT TEMPLATE FOR INTENT DECOMPOSE.

Here we provide the Detailed Prompt Template for INTENT DECOMPOSE.

### A.1.1    INTENT CONSTRAINT GENERATION

Table 5 provides the detailed prompt of Intent Constraint Generation in INTENT DECOMPOSE. We put all steps together instead of seperating them for (1) efficiency, one call of LLM is enough and (2) self-consistency, user may run this prompt for multiple times to ensure the constraint consistency.

| Component | Details |
|---|---|
| **Prefix** | You are an advanced linguist tasked with processing queries using a constraint-based approach. Decompose the given query step by step, following the instructions below.

`Query:` **Existing Content** |
| **Suffix** | **0. Preliminary Check:**
    - Focus solely on the TASK QUERY.
    - Check if any external content, documents, or data are provided.
    - Verify if ALL NECESSARY external contents are provided.
If ANYTHING is missing, request clarification.
Example: If the user asks you to evaluate a response based on a given article but forgets to provide it, you should request the missing information.
**If the Preliminary Check fails,** IGNORE the following steps and politely ask for clarification. Use "START:" to begin the final listing. |
| | **1. Identify Core Elements:**
    - Determine the main subject, action, and context of the query. Focus on the query's intent, but not the task itself (e.g., put words like "name/list" as an action).
    - Ensure the necessary content is available if the action involves processing external content.
    - DECOMPOSE AS THOROUGHLY AS YOU CAN. EACH ELEMENT MUST BE A SINGLE OBJECT, NOT MULTIPLE. Do not overanalyze the query—if the query is simple, then it would not have many constraints. |
| | **2. Decompose into Constraints:**
**a) Essential Components Extraction:**
    - Identify all explicit conditions, requirements, or limitations in the query.
    - Map each to one of the following components: Location, Time, Subject, Action, Qualifiers, Quantity.
    - Treat each condition as a separate constraint.
**b) Constraint Prioritization and Formulation:**
    - For each constraint, assess its importance:
        - **Mandatory**: Critical elements that must be addressed.
        - **Important**: Elements that should be addressed if possible.
        - **Optional**: Elements that can be addressed if convenient.
        - Formulate constraints for each component, specifying the priority, using the template:
    "[Priority Level]: [Component] must/should [condition]"
**At the end,** provide the list of constraints a response should cover, grouped by priority levels ONLY. Use "START:" to begin the final listing.
YOU MUST ONLY LIST THE FINAL CONSTRAINTS AT THE END, AFTER START. NOTHING ELSE. |

Table 5: The final prompt is Prefix + `Query` + Suffix.

### A.1.2 CONSTRAINT SCORE

### A.1.3 FACT CHECK

We manually checked the performance of self-consistency over 100 cases with GPT-4o under $\tau = 0.3$. We found that for 93 cases the results are consistent and accurate, indicating it is providing the correct outcome. For the rest 7 cases, the 5 false-factual-inaccurate cases are detected by LLMs, leaving only 2 wrong cases. Due to monetary constraint and time constraint, we believe this result is satisfying enough for us to adopt Self-Consistency method.

### A.2 AUTOMATIC CONSTRUCTION PIPELINE FOR FAITHQA

As the setups of **Omission** and **Misinterpretation** are different, we designed different generation pipelines tailoring each dataset.

| Datasets | | | FAITHQA: Dataset Statistics | | |
|---|---|---|---|---|---|
| | | | **Easy** | **Hard** | **Total** |
| **Minor Fabrication** | | | | | |
| FactQA | Open Answer | Tech | 500 | 500 | 1000 |
| | | Culture | 500 | 500 | 1000 |
| | | History | 500 | 500 | 1000 |
| Creative Writing | Story | – | 500 | 500 | 1000 |
| | Poem | – | 500 | 500 | 1000 |
| **Major Fabrication** | | | | | |
| Response Evaluation | | Tech | – | – | 810 |
| | | Health | – | – | 750 |
| | | Culture | – | – | 810 |
| | | History | – | – | 840 |
| Content Analysis | Relationship | Tech | – | – | 1431 |
| | | Health | – | – | 1225 |
| | | Culture | – | – | 1436 |
| | | History | – | – | 1837 |
| | Summary | Tech | – | – | 1431 |
| | | Health | – | – | 1225 |
| | | Culture | – | – | 1436 |
| | | History | – | – | 1837 |

Table 6: Dataset statistics for FAITHQA. Each cell shows the number of problems across difficulty and topic. Easy: constraints $\leq 4$, Hard: constraints $> 4$.

### A.2.1 GENERATION PIPELINE FOR MINOR FABRICATION.

We utilized GPT-4o to sample for the problems, by manually giving GPT-4o exemplar questions we created. GPT-4o is able to transfer among the topics and adjust to different cinstraint amounts by providing different exemplars.

### A.2.2 GENERATION PIPELINE FOR MAJOR FABRICATION.

**Major Fabrication** is a RAG dataset, therefore we first sampled 50 articles for each topic to start from. We then composed 3 pairs of (query, response) for each article.

### A.2.3 QUALITY CONTROL

After acquiring the initial dataset, we carried out a comprehensive data cleaning and quality assessment process. This included a manual review of each example to ensure that the questions were well-constructed, removing any duplicates and eliminating invalid questions (such as those that were overly simple or potentially controversial).

