# OpenReview forum: "Instruction Following is not all you need: Rethinking LLM Generation's Evaluation"
_ICLR.cc/2025/Conference — ICLR 2025 Conference Withdrawn Submission_

### Official Review · Reviewer_M97y · 2024-10-20

**Soundness:** 2
**Presentation:** 1
**Contribution:** 2
**Rating:** 3
**Confidence:** 4

**Summary:**

The focuses on the problem of intent hallucination. where a LLM follows the instruction but contain factually incorrect statements. This paper first introduces how to measure the intent hallucination ratio of several LLMs (Intent Decompose). Second, the paper introduces FaithQA, which by dividing the measure of intent hallucination into two parts: non-paragraph level minor fabrication, and paragraph level major fabrication.

**Strengths:**

- The paper tackles an important problem of LLM evaluation which is referred to as 'intent hallucination' in this work.
- The paper explores various LLMs in the Experiments section.

**Weaknesses:**

- The writing of the paper is not clear and hinders readability.
- Many details are missing regarding the FaithQA benchmark construction process. For example, how is the quality of the dataset controlled in detail? (the appendix mentions that "manual review of each example to ensure that the questions were well-constructed" which is very vague making the quality of the benchmark questionable)
- In the introduction, it is mentioned that "We perform human evaluation to prove the effectiveness of INTENT DECOMPOSE in detecting and quantifying intent hallucination" in line 113-114. However, human evaluation of intent decompose does not exist in the paper. This makes the reliability of the result of Table 1 and 2 questionable.
- The contribution of the proposed Intent Decompose is incremental compared to existing works for fine-grained LLM evaluation (Min et al 2023, Ye et al 2024).

References:

[1] Min et al; FActScore: Fine-grained Atomic Evaluation of Factual Precision in Long Form Text Generation (EMNLP 2023)

[2] Ye et al; FLASK: Fine-grained Language Model Evaluation based on Alignment Skill Sets (ICLR 2024)

**Questions:**

- What are the ratio for each error case for the result of Section 7 (analysis part)?
- The paper mentions that the authors evaluate LLMs and various prompting techniques on the test set of FaithQA due to monetary costs. How large is the test set of FaithQA?

---

### Official Review · Reviewer_6Uao · 2024-11-02

**Soundness:** 1
**Presentation:** 2
**Contribution:** 1
**Rating:** 3
**Confidence:** 4

**Summary:**

This paper proposes an issue called *intent hallucination*, highlighting the phenomenon where LLMs follow the instruction but produce unfactual outputs. To address this, the authors introduce a benchmark called FaithQA and a corresponding detection method named *intent decompose* to identify response hallucination.

**Strengths:**

- The intent to break down instruction-following and content-factuality is a meaningful guideline for designing LLM benchmarks.
- The idea to decompose the query and make point-wise judgments is excellent for LLM evaluation.

**Weaknesses:**

1. The story of this paper is partly unconvincing to me. I believe there are many open-ended generative datasets (e.g., Natural Questions and AlpacaEval) that don't involve difficult instruction following. Researchers can directly use them to identify hallucinations in generated texts.

2. The paper lacks many important details.
   - The construction of the FaithQA dataset is completely missing. What is FactQA? Is it self-built or existing data? Where do the queries come from? What is the query format? Where do the external contents come from? How do you conduct extensive human evaluations?
   - The weight of constraint score is missing. The meaning of these weights is unexplored.

3. The experiment is not robust.
   - The premise that lower identification rates indicate better methods is entirely unconvincing. You should add human evaluation as a benchmark. The method closest to human evaluation has the best performance.
   - Only one baseline method is used. If you want to highlight the effectiveness of your decomposition method, you should include more baseline methods.
   - The analysis is not convincing. How do you draw your four conclusions? At the very least, you need to quantitatively analyze the proportion of these situations, rather than relying on a few examples.

4. Your query decomposition method needs further exploration. Do you verify whether LLM can truly break down queries into your given categories, and how do you do so?

**Questions:**

See weakness for details.

There also exist several typos or expressions that need to be improved:
- line 38, a instruct -> an instruct
- line 39 and many others, the left quota should use `` in latex.
- line 45 and many others, requirements(Liu -> requirements (Liu
- line 138, missing verb.
- line 153, the expression should be improved as "we define instruction following as whether LLMs address word-level concepts or actions."
- line 229 and many others, R=P(M|q) is a wrong equation.
- line 475 and many others, you should check all the citation and correctly use `citep` and `citet`,

---

### Official Review · Reviewer_U76q · 2024-11-04

**Soundness:** 2
**Presentation:** 2
**Contribution:** 2
**Rating:** 3
**Confidence:** 4

**Summary:**

This paper proposes an evaluation benchmark called FaithQA, which focuses on evaluating the intent hallucination problem of LLMs. The work constructed more than 18k evaluation samples, separated into non-paragraph level minor fabrication and paragraph level major fabrication subsets. For each subset, the evaluation query is paired with fine-grained importance-weighted intents. The minor fabrication subset is aimed at the model's factually correctness for specific world knowledge, while the major fabrication part tests the model's awareness of key context information missing in the given query. During evaluation, the satisfaction of query intents is automatically judged by judger model or Wikipedia, deriving the overall score on FaithQA. Various open-sourced or proprietary LLMs are evaluated and compared on this benchmark.

**Strengths:**

1. The motivation of the paper is reasonable. Current evaluation benchmarks may usually underweights the importance of factual correctness of the models.
2. The size of FaithQA is large, containing more than 18k problems, each paired with fine-grained weights.
3. Both open-sourced or proprietary LLMs are evaluated and compared on this benchmark.

**Weaknesses:**

1. Significant lacking of details on this benchmark, with no samples in the supplementary details. At least one sample from each subset of minor fabrication (like Tech domain) and major fabrication (like Relationship subtask) should be given.
2. Weird presentation error in the main experimental results: for the accuracy of Mistral-7B in minor fabrication subset, the results in Table 1 and Table 2 are hugely inconsistent (like Culture domain, in Table 1 the Acc. is 0.63, however in Table 2 it is 0.15 and 0.09 on Easy and Hard subsets, respectively).
3. The details of conducting the automatic evaluation are so brief. At least the prompt of LLM-as-a-judge should be provided. Moreover, how does FaithQA determines the importance of each intent? I do not see any section discussing this.
4. No insightful analysis appears in the paper.
5. No correlation analysis between human evaluation and automatic evaluation of FaithQA on these LLMs are provided.

**Questions:**

None

---

### Official Review · Reviewer_zpa7 · 2024-11-10

**Soundness:** 3
**Presentation:** 1
**Contribution:** 2
**Rating:** 3
**Confidence:** 4

**Summary:**

This paper proposed FAITHQA, a novel benchmark for intent hallucination that contains 18,068 queries, covering both query-only and retrieval augmented generation (RAG) setups with varying topics and difficulty. FAITHQA aims to evaluate LLMs' intent hallucination problem which can manifest in two granulated ways: minor fabrication, where the response introduces sentence-level factually incorrect information or major fabrication, where the paragraph level of the response is entirely factually inaccurate or fabricated. By evaluating SOTA LLMs on this benchmark, the authors showed that  intent hallucination appears across different model families and sizes of LLMs. Finally, the authors introduced INTENT DECOMPOSE, a novel approach for detect intent hallucination. This method evaluates LLM generations based on breaking query into intent constraints and compute a weighted score. Through human evaluations, they showed the effectiveness of INTENT DECOMPOSE in detecting and quantifying intent hallucination.

**Strengths:**

- This paper is technically sound and proposes a benchmark for evaluating intent hallucination. The idea is generally valid but many parts of the paper are very hard to understand. See weakness for details.

**Weaknesses:**

- This paper missed a large part of RAG evaluation literature. For example, [Chen et al., 2024](https://arxiv.org/abs/2309.01431) evaluated 4 fundamental abilities required for generators; [Ru et al., 2024](https://arxiv.org/pdf/2408.08067) introduced RAGChecker, a method to evaluated RAG system in a comprehensive way by decomposition; [Liu et al.,2024](https://arxiv.org/pdf/2311.08147) introduced manually edited counterfactual contexts into QA and text generation datasets to evaluate the counterfactual robustness of LLMs. How does this work compared to those as it's also largely focused on hallucination in RAG settings.

- I am quite confused by section 4.1 which introduces the construction of the FAITHQA benchmark: 1) what is the "multi-constraint FactQA setup" mentioned in Line 276? It's the first time where "FactQA" appears in the paper (correct me if I am wrong) but without a citation. Also, there is no example provided in the paper to show what a real example looks like in the paper. 2) For Major Fabrications in Line 286, how is RAG setup actually adapted to construction the benchmark? 3) The evaluation metric and the evaluation protocol are also unclear to me. In general, I don't have a good judgment of the benchmark quality.

- Section 5 is also confusing. How is the Acc computed in Table 1 and why there is a large gap between base and Acc? If both base and acc captures the intent hallucination rate, why is it listed in the table like two metrics?

**Questions:**

See weakness section.

---

### Note · Authors · 2024-11-25

**Comment:**

We sincerely appreciate the useful advices from the reviewer. We here decide to withdraw our submission to further refine our draft. Thank you so much for the precious time and effort.

**Withdrawal Confirmation:**

I have read and agree with the venue's withdrawal policy on behalf of myself and my co-authors.